# The Longitudinal Association between Co-Residential Care Provision and Healthcare Use among the Portuguese Population Aged 50 and Over: A SHARE Study

**DOI:** 10.3390/ijerph20053975

**Published:** 2023-02-23

**Authors:** Fátima Barbosa, Sara Simões Dias, Gina Voss, Alice Delerue Matos

**Affiliations:** 1Communication and Society Research Centre, Institute of Social Sciences, University of Minho, 4710-057 Braga, Portugal; 2Center for Innovative Care and Health Technology (CiTechCare), School of Health Sciences, Polytechnic of Leiria, 2410-541 Leiria, Portugal; 3Department of Sociology, Institute of Social Sciences, University of Minho, 4710-057 Braga, Portugal

**Keywords:** co-residential care, number of visits to the doctor, Portugal, SHARE, longitudinal analysis

## Abstract

Co-residential care is associated with poor caregiver health and a high burden. Although Portugal relies heavily on co-residential care by individuals aged 50 and over, studies on the impact of co-residential care provision on Portuguese caregivers’ healthcare use are lacking. This study aims to analyze the impact of co-residential care (spousal and non-spousal care) on healthcare use of the Portuguese population aged 50 plus. Data from waves 4 (*n* = 1697) and 6 (*n* = 1460) of the Survey of Health, Ageing and Retirement in Europe (SHARE) were used. Negative Binomial Generalized Linear Mixed Models with random (individual level) and fixed (covariates) effects were performed. The results show that the number of visits to the doctor decrease significantly over time for the co-residential spousal caregivers as compared to the non-co-residential caregivers. This result highlights the fact that the Portuguese co-residential spousal caregiver group is at a higher risk of not using healthcare, thus jeopardizing their own health and continuity of care. Promoting more accessible healthcare services and implementing public policies adjusted to the needs of informal caregivers are important to improve the health and healthcare use of Portuguese spousal co-residential caregivers.

## 1. Introduction

Population ageing represents a significant challenge for long-term care systems [1]. In Europe, informal care is the primary source of long-term care (LTC) [2,3], with family providing 80% of all LTC [4,5]. Informal caregivers are crucial for promoting the well-being of people who need care and stimulating countries’ economies. [6]. Although informal care is regarded as an alternative to healthcare services and institutional care [2,7], several studies have shown that informal care impacts informal caregivers’ health and healthcare utilization. The research also indicates that informal caregivers should be viewed as secondary patients [8,9] and that promoting their health is vital to ensuring the sustainability of care systems and the continuity of informal care provision [6,7].

The literature points out that the association between informal care provision and caregiver health is influenced by the context (out-of-home care and co-residential care) in which informal care is provided, the relationship to the care recipient (spouse, parents, etc.), and the European region [10,11,12]. More specifically, caregivers providing informal care within the household (co-residential care) tend to report poorer health than their peers who provide informal care outside the household [1,10]. The main reasons for this are: advanced age, poorer physical and mental health, higher levels of stress and burden, emotional closeness to the person being cared for, and the fact that co-residential caregivers perform more burdensome care tasks that compete with other time-intensive activities [1,10,13,14,15]. Regarding the relationship to the care recipient, spouse caregivers report more depressive symptoms than child caregivers [16,17]. They are usually older [18], provide more hours of care, have less time to rest from their caring role [19], and usually have difficulty recognizing their needs [20]. All of these are factors that may negatively affect their health. Lastly, informal caregivers in Southern and Eastern Europe tend to provide more intensive care, have less formal support [21], and experience worse health conditions than informal caregivers in Northern Europe [22].

The available research into the relationship between informal care provision and healthcare use by informal caregivers shows mixed results. These studies also use different measures of healthcare use assessment. A recent study [23] drew attention to the need to identify which characteristics of caregivers lead to more significant inadequate healthcare utilization; for example, the place of care and the type of relationship between caregiver and care recipient. Overall, some studies showed that informal caregivers access more healthcare than non-caregivers [19,24,25,26,27]. The high stress and anxiety experienced, poor health (higher levels of depression, lower self-perceived health, and chronic illness), limited rest and exercise time, and few preventive behaviours were determining factors that contribute to higher healthcare use by informal caregivers as compared to non-caregivers [19,24,26]. Close proximity to the health problems of the person cared for can also lead to a greater awareness of health problems and higher healthcare use by the informal caregiver [19]. On the other hand, other studies [28,29] found that informal caregivers are at a higher risk of not seeking medical care due to lack of social support, financial problems, and long-term illness [28,29]. Other studies [23,30] also showed that informal caregivers and non-caregivers do not differ in terms of the number of medical consultations, routine check-ups, inpatient admissions, or instances of primary care. The authors [30] invoked the healthy caregiver hypothesis [11,31], which argues that informal caregivers are healthy people and that providing informal care does not interfere with informal caregivers’ healthcare utilization.

The OECD data show that Portugal is the fourth country with the highest percentage of the population aged 65 and over, with projections indicating a doubling of the population aged 80 and over between 2019 (6.4%) and 2050 (12.8%) [32]. Several studies also show that Portugal has the largest percentage of co-residential caregivers aged 50 and over [33,34] and of intensive caregivers (who provide 11 hours or more of care/help per week) [21]. In addition, in Portugal, the family continues to be the most important provider of care for older and dependent people [35,36,37,38], and Portuguese people aged 50 and over have higher unmet social care needs [38]. Despite this, the level of health expenditure per capita is low and public provision of community care services in Portugal is limited [32,39,40]. Although the Portuguese National Health Service provides all services (except dental care) to the entire population, inequalities in accessing healthcare in Portugal are well-documented [41,42,43,44,45,46,47]. The economic crisis and subsequent implementation of austerity measures in 2010 had a negative impact on the Portuguese National Health System and made it more difficult to access healthcare [41,44,48,49]. Nevertheless, in the last decade, Portuguese civil society and policymakers have made great efforts to support informal caregivers. In 2019, the Portuguese government approved the formal status of informal caregivers [50,51]. This includes a monetary subsidy for the caregiver, caregiver respite services, psychosocial support, and measures to support integration into the labour market [52]. Although this status is essential in the current Portuguese context, implementation is still at a very early stage.

Given that Portugal relies heavily on co-residential care provision by individuals aged 50 and over, our aim was to conduct the first longitudinal study analyzing the association between the provision of co-residential care and the use of healthcare among the Portuguese population aged 50 years or older. Specifically, we compared healthcare utilization by the co-residential spousal caregiver group and the non-spousal co-residential caregiver group with the non-caregiver group over time (between wave 4 and wave 6). As we focused on the middle-aged and older population—those most at risk of experiencing health problems—and informal caregivers usually provide care for long periods of time [53,54], longitudinal analysis was required to determine whether co-residential informal care provision over time limits or enhances healthcare use by Portuguese individuals aged 50 and over. Lastly, our study aims to contribute to the current literature on informal caregiver healthcare use by focusing on Portugal (the country with the highest percentage of co-residential caregivers in Europe) and on co-residential care provision, and also by specifying the relationship between caregiver and care recipient (spousal and non-spousal).

Taking into account existing Portuguese vulnerabilities in terms of healthcare access, long-term care, and formal support to informal caregivers, as well as the vulnerabilities of the spousal caregiver group (provision of many hours of care, limited time to rest, and difficulty recognizing their own needs) compared with the non-caregiver group, we hypothesized that the Portuguese spousal caregiver group experiences a significant decrease in the number of visits to the doctor over time.

## 2. Materials and Methods

### 2.1. Study Population

This longitudinal study uses data from waves 4 and 6 of the SHARE (Survey of Health, Ageing and Retirement in Europe) project, version 7.1.0 (https://doi.org/10.6103/SHARE.w4.710; https://doi.org/10.6103/SHARE.w6.710). SHARE is a multidisciplinary, multi-national study that includes representative samples of respondents aged 50 and over and their partners, regardless of age, from 27 European countries and Israel.

Since 2004, SHARE has collected data every two years, providing internationally comparable longitudinal micro-data that offer a perspective on European individuals’ public health and socio-economic living conditions (http://www.share-project.org/, accessed on 1 December 2022). Portugal was part of the SHARE study in wave 4 (2011) and to the present date has conducted: wave 4 (2011); wave 6 (2015); wave 7 (2017–2018—Life Stories); COVID-19 survey wave 8 (2020); COVID-19 survey wave 9 (2021); and wave 9 (2021–2022). For more methodological details of the SHARE project, please see Börsch-Supan et al. [55].

With regard to Portugal, the variables of interest are only available in waves 4, 6, and 9, and the data from wave 9 are not yet available. We therefore restricted our sample to Portuguese people aged 50 and over who participated in waves 4 (*n* = 1697) and 6 (*n* = 1460). Figure 1 displays the flowchart outlining our study population in the two waves analyzed. The flowchart shows that 1697 individuals aged 50 plus completed the SHARE Portuguese Questionnaire in wave 4, 1451 in wave 6, and 1219 individuals participated in both waves analyzed.

### 2.2. Measures

Number of visits to the doctor in the last 12 months is a widely used measure to assess healthcare use [56,57,58]. In wave 4, the variable number of visits to the doctor included the total number of visits to the doctor in the last 12 months, excluding visits to dentists and hospital stays, but including visits to emergency or outpatient clinics. In wave 6, this variable also covered a qualified/registered nurse appointment. For both variables, the values range from 0 visits to 98 visits to the doctor in the last 12 months.

Providing co-residential care is the variable of interest. Only those not living alone were asked whether they provided regular (daily or almost daily for at least three months) care (washing, getting out of bed, or dressing) to someone living in the same household. If the answer was yes, the respondents were asked to whom they provide care. This variable was categorized into three groups: non-co-residential caregivers (if the respondent answered no to the first question); co-residential spousal caregivers (if the respondent answered yes to the first question and reported in the second question caring for the spouse living with him/her); and non-spousal co-residential caregivers (if the respondent answered yes to the first question and reported in the second question caring for other people living with him/her but not for their spouse).

Based on the literature, a set of covariates were included in the model. Age at the time of the interview and gender were inserted as control variables. The employment status variable was categorized into retired and other categories (employed, unemployed, permanently ill or disabled, homemaker, or other). The level of education was assessed through the International Standard Classification of Education (ISCED 97), which was grouped into three levels: low education level (individuals who have not attended school or have only completed primary education or completed the third cycle); medium education level (individuals who have completed secondary education); and high education level (post-secondary education, but not tertiary education or higher education). Financial distress was analyzed through the following question: Thinking of your household’s total monthly income, would you say that your household is able to make ends meet: 1. With great difficulty; 2. With some difficulty; 3. Fairly easily; 4. Easily. This question was assessed using a dichotomous variable, with options 1 and 2 representing the presence of financial stress (1) and options 3 and 4 representing the absence of financial stress (0).

Regarding health, the number of chronic diseases and having been hospitalized in the last 12 months were also considered. We used the EURO-D scale with 12 items [59] to assess depression. This scale assesses feelings of depression, pessimism, death wish, guilt, irritability, crying, fatigue, sleep problems, loss of interest, loss of appetite, reduced concentration, and loss of enjoyment of life during the past month. The EURO-D scale ranges from 0 (not depressed) to 12 (very depressed) and, according to Dewey and Prince [60], a score equal to 4 or more indicates clinical depression. Thus, respondents with 4 or more symptoms were coded as 1 (clinical depression) and the others with 0 (not clinical depression). Physical inactivity was measured according to the procedures in Gomes et al. [61]. In the SHARE study, respondents were first asked how often in their daily lives they engaged in vigorous activity (i.e., sports, heavy housework, or a job that requires physical work) and then how often they engaged in moderate activity (i.e., activities that require a low or moderate energy level, such as gardening, cleaning the car, or walking), with four response options: 1. more than once a week; 2. once a week; 3. one to three times a month; 4. hardly ever, or never. Individuals who answered “one to three times a month” and “almost never, or never” to both questions were considered physically inactive (physically active = 0 and physically inactive = 1). The total number of people living in the household and the total number of activities carried out by the respondents in the last 12 months were also included. In wave 4, the SHARE questionnaire included the following activities: done voluntary or charity work; attended an educational or training course; gone to a sports, social or other kind of club; taken part in activities of a religious organization (church, synagogue, mosque etc.); taken part in a political or community-related organization; read books, magazines or newspapers; did word or number games such as crossword puzzles or Sudoku; played cards or games such as chess; and none of these. In wave 6, the question remained the same; however, option 4 (taken part in activities of a religious organization (church, synagogue, mosque etc.) was not considered. Finally, time was coded as 1 (wave 4) and 2 (wave 6).

### 2.3. Statistical Analyses

We first performed a descriptive analysis of all selected variables of waves 4 and 6. Since the health and economic variables had missing values greater than 5% [62], the imputations provided by the SHARE study [63] were applied to such cases. Secondly, we examined the socio-demographic, economic, and health differences between the three analyzed groups (non-co-residential caregivers; co-residential spousal caregivers; and non-spousal co-residential caregivers) using the ANOVA (*F*) and Tukey’s post hoc tests for continuous variables and the chi-square test (X^2^) for categorical ones. As the SHARE sample does not have a uniform design, we used calibrated individual weights only for percentages and means. Thirdly, Generalized Linear Mixed Models for count data were applied to analyze the longitudinal association between co-residential care provision and healthcare use among the Portuguese population aged 50 and over. The Generalized Linear Mixed Models are an extension of the Generalized Linear Models, allowing correlation between observations through the insertion of random effects. The systematic component integrates fixed effects and random effects, which makes it possible to analyze data with correlated observations; that is, data repeated over time for the same individual:(1)$$\eta_{ij}=\beta_{0}+\beta_{1}\times X_{1ij}+\beta_{2}\times X_{2ij}+\ldots +\beta_{p}\times X_{pij}+\alpha_{i}$$
(2) i=1,…, n     j=1, 2
where $\alpha_{i}∼N\left(0,\sigma_{\alpha }^{2}\right)$ is the random effect.

In this study, Generalized Linear Mixed Models with random (individual level) and fixed (covariates) effects were used. As the Generalized Linear Mixed Models with Poisson response indicated the presence of overdispersion ($\hat{\emptyset }$ = 1.282; *p* < 0.001), Generalized Linear Mixed Models with Negative Binomial response were conducted [64,65,66].

In this sense, four Generalized Linear Mixed Models with Negative Binomial response using the maximum likelihood estimation, Laplace approximation, were analyzed:−Null Model (only included dependent variable: number of visits to the doctor in the last 12 months);−Model 1 (included, in addition to the variable inserted in Null Model, the interest variable providing co-residential care);−Model 2 (included, in addition to the variables inserted in Model 1, the following variables: age, sex, current job situation, education, financial distress, hospital admissions, number of chronic diseases, depression, physical inactivity, household size, social activities and time (wave));−Model 3 (included, in addition to the variables inserted in Model 2, the interaction term between providing co-residential care and time).

Appendix A presents the statistical values of the Variance Inflation Factor (VIF) and Tolerance for Model 2. Considering that all VIF values are smaller than 5 (VIF < 5) and Tolerance values are higher than 0.20, there are no problems of multicollinearity [67]. Incidence Rate Ratios (IRR) with 95%CI were computed. Goodness-of-fit measures (Akaike Information Criterion (AIC), Deviance and Log-Likelihood) were used to compare different models.

The analyses were conducted in R 4.0.2 software, library nlme, and the significance level of *p* established was 0.05.

## 3. Results

### 3.1. Descriptive Statistics

Table 1 shows the descriptive statistics of Portuguese aged 50 plus at baseline (wave 4), according to caregiver status. The data show that at baseline (wave 4), in comparison with the other analyzed groups, the co-residential spousal caregivers group was older (68.6, compared with 65.9 for the non-co-residential caregiver group and 66.3 for non-spousal co-residential caregivers), predominantly male (58.8%, compared to 48% of the non-co-residential caregiver group and 21% of non-spousal co-residential caregivers) and retired (75.6%, compared to 57.7% of non-spousal co-residential caregivers and 50.9% of the non-co-residential caregiver group). Concerning health, the group of non-spousal co-residential caregivers had a higher number of chronic diseases (2.5) compared to the co-residential spousal caregiver group (2.3) and non-co-residential caregiver group (1.8). Lastly, the co-residential spousal caregiver group reported higher percentages of four or more depressive symptoms (64.3%, compared to 52.8% of the group of non-spousal co-residential caregivers and 40.7% of the non-co-residential caregiver group) and a lower average number of people living in the household (2.6 compared with 2.7 of the non-co-residential caregiver group and 3.2 of the group of non-spousal co-residential caregivers). At baseline, no significant differences were found between the analyzed groups in terms of education (*p*  =  0.466), financial stress (*p*  =  0.069), number of hospitalizations in the last 12 months (*p*  =  0.805), physical inactivity (*p*  =  0.895), social activities (*p*  =  0.838), or doctor visits in the last 12 months (*p*  =  0.154).

### 3.2. Longitudinal Analysis

Table 2 shows the longitudinal association between co-residential care provision and healthcare use among the Portuguese population aged 50 and over. The results indicate that, considering the goodness-of fit-measures, Model 3 (in Table 2) presents a better fit than the other models analyzed. Therefore, Model 3 shows that, at baseline (wave 4), for a significance level of 5%, providing co-residential care (co-residential spousal care (IRR:1.33, 95%CI: 0.99–1.79) or co-residential non-spousal care (IRR:0.89, 95%CI: 0.66–1.21)) is not associated with number of visits to the doctor in the last 12 months. Nevertheless, the interaction term between providing spouse co-residential care and time shows that over time (between wave 4 and wave 6), the group of co-residential spousal caregivers, compared to the non-co-residential caregiver group, was associated with a significant decrease (IRR:0.86, 95%CI: 0.76–0.98) in the number of visits to the doctor in the last 12 months. No statistically significant differences were found between providing non-spousal co-residential care and time (IRR:1.06, 95%CI: 0.92–1.21) regarding the number of visits to the doctor in the last 12 months. These results can be better observed in Figure 2, which shows that, between wave 4 and wave 6 the group of co-residential spousal caregivers, compared to the non-caregiver group (1.67 in wave 4 and 1.94 in wave 6), is associated with a statistically significant decline in the number of visits to the doctor—from 1.23 in wave 4 to 1.06 in wave 6. No statistically significant differences were found in the number of visits to the doctor by co-residential non-spousal caregivers (1.86 in wave 4 and 2.40 in wave 6) compared with non-co-residential caregivers (1.67 in wave 4 and 1.94 in wave 6).

Model 3 further shows that having a higher educational level (IRR:1.24, 95%CI: 1.09–1.42) was associated with a 24% increase in the number of visits to the doctor. Along the same lines, individuals who reported having been hospitalized in the last 12 months (IRR:1.72, 95%CI: 1.56–1.89), those who reported four or more depressive symptoms (IRR:1.26, 95%CI: 1.17–1.35), and those who were physically inactive (IRR:1.16, 95%CI: 1.06–1.26) were associated with an increase of 72%, 26%, and 16%, respectively, in the number of visits to the doctor. Furthermore, each additional chronic disease reported is associated with 14% (IRR:1.14, 95%CI: 1.12–1.16) more visits to the doctor. Lastly, each additional time (wave) is associated with an increase of 8% (IRR: 1.08, 95%CI: 1.04–1.11) in the number of visits to the doctor. The interaction term between providing co-residential care and gender was also tested (Model 4 in Appendix A). As the interaction was not significant and negatively affected the model, it was not considered in this study.

## 4. Discussion

In a context of population ageing, there tends to be an increase in informal caregiving by middle-aged and older adults. As Portugal is the European country with the highest percentage of informal co-residential caregivers aged 50 plus, and co-residential care provision is associated with poor caregiver health and a high burden, it is extremely important to assess the relationship between co-residential informal care provision and the use of healthcare by Portuguese individuals aged 50 years and over.

Our analysis showed that, across the analyzed waves, compared with the non-co-residential caregiver group, Portuguese co-residential spousal caregivers showed a decrease in the number of visits to the doctor. These results validate our research hypothesis that this group is at a higher risk of not using healthcare. Our results point in a different direction from the findings by Zwart et al. [19], which showed that European female co-residential spousal caregivers from 11 European countries (not including Portugal) showed an increase in healthcare usage over time. A number of differences in the study design and also the Portuguese context can help to explain our results. The study by Zwart et al. [19] focused on co-residential care provision, has a longitudinal perspective, and used the same measure of healthcare use (number of visits to the doctor in the last 12 months). However, it only focused on spousal care provision, did not include Portugal, did not conduct a country analysis (the analyses were adjusted by European regions: North, Centre, and South), and used different statistical techniques (statistical matching, whereas we used mixed-effects regressions).

Furthermore, it is well established that in Portugal the family is the main provider of care [37,68] and public policies to support informal caregivers are still residual [32,39]. Furthermore, even though everyone can have access to healthcare through the Portuguese National Health System, inequalities in access are extremely widespread [42,44,47,69,70]. Doetsch et al. [46] highlight the existence of several barriers to healthcare access for Portuguese older people. Portugal is also the Southern European country where people aged 50 and over have the greatest need for care and where people are most likely to receive only informal care [38]. The ideology of familialism is based on instrumentality, the existence of weak formal support, and a model in which individuals have learned to be self-reliant [35,71]. Since informal support is often considered normal in Portugal due to established social norms and family and social arrangements [51], spousal care may seem like a natural part of the intimate relationship [53] and prevent spouse caregivers from asking for help. As long-term care services in Portugal are underdeveloped [32], spousal caregivers also lack options, and may feel pressured to provide care regardless of their advanced age or poor physical and mental health [72]. In this sense, despite the fact that caring for a dependent spouse is very challenging, many older spouses prefer to continue providing care and only cease to be informal caregivers when they are no longer capable of doing so [72]. Our findings also align with a previous study which found that lack of social support and informal care provision are associated with lower medical care utilization [28]. In this sense, the limited support received, the lack of options, and the fragilities of social care and healthcare in Portugal may lead Portuguese spouse caregivers to experience difficulties in accessing healthcare services as they prioritize the dependent spouse’s needs to the detriment of their own. The findings of our study reinforce a recent study [38] which stressed that Portugal performs poorly compared to other Southern European countries in terms of meeting the long-term care needs of the population aged 50 and over.

Spouse caregivers are considered a vulnerable group as they are older, have poor health [18], are often solo caregivers, and usually perform intensive and burdensome care (provide a higher number of hours of care, have less informal and formal support, and have less time to rest) [16,18,19,53,72,73,74]. The intensive care provided and lack of formal support in Portugal contribute to an increased informal care burden and a decline in spouse caregiver health. This prevents them from accessing and seeking healthcare due to a lack of time and their own poor health. Spousal caregivers also have difficulty identifying and recognizing their own needs, which results in underestimation of their unmet needs [20]. Older Portuguese people also have lower levels of health literacy [75]. In addition to their advanced ages [76], this explains why Portuguese spouse caregivers experience difficulty assessing their health status and accessing healthcare services [77]. As they may not acknowledge their role as informal caregivers, they will therefore not ask for support and also neglect their own health.

It is also important to stress that this study covered the period of the economic crisis in Portugal, a time when most of the population experienced a decrease in income. This resulted in more unmet health needs [45] and reduced access to healthcare [46,78], especially among the older age group [46,78]. This may also have impacted access to healthcare by older spousal caregivers.

Our study also found no differences between genders in terms of the number of visits to the doctor by co-residential caregivers (spousal or non-spousal) as compared to non-co-residential caregivers. Our findings do not support the results cited by Zwart et al. [19], which showed that healthcare usage by female spouse caregivers increased over time. First, this result can be explained by the fact that Zwart et al. [19] focused on a different population (individuals aged 50 and over from 11 European countries, not including Portugal). Second, male and female spouse caregivers provide similar care hours [79] and, in countries with less formal support, such as Portugal, these gender differences may not exist among spouse caregivers as they have less time for self-care and less access to healthcare services. In addition, in Portugal no gender differences in health literacy were found [75].

To our knowledge, this is the first study analyzing the longitudinal association between providing co-residential care and healthcare usage in the Portuguese population aged 50 plus. This study aims to contribute to the literature by focusing on co-residential care provision in Portugal and by differentiating between two groups of co-residential care providers: spousal caregivers and non-spousal caregivers. Nevertheless, this study has limitations. The dependent variable number of visits to the doctor in the last 12 months is based on memory and may be subject to underreporting (memory bias). Furthermore, this question underwent some changes between waves, which may have affected the analysis (while the wave 4 SHARE questionnaire only asks if the respondent has seen or talked to a medical doctor about his/her health, in wave 6 the questionnaire asks if the respondent has seen or talked to a medical doctor or qualified nurse about his/her health). Because Portugal only joined the SHARE study at wave 4 and our variables of interest are only available for waves 4 and 6, our analysis only covered two waves. In addition, the SHARE study does not indicate the number of hours of informal care provided by the informal caregivers, which prevented us from analyzing the intensity of care provided by informal caregivers.

Given the different levels of development of social and health systems and different intensities of informal care provision in European countries, further research into healthcare use among informal caregivers should adopt a country perspective. It should also consider the place where the care is provided and the type of caregiver/care recipients relationship. The long-term impacts of informal caregiving on caregiver healthcare use, including after the death of the person cared for, can also help us better understand the real impact of informal caregiving on caregiver health and on healthcare use.

## 5. Conclusions

Our study showed that between wave 4 (2011) and wave 6 (2015), compared to the non-caregiver group, the Portuguese co-residential spouse caregiver group showed a statistically significant decrease in the number of visits to the doctor. The decrease may indicate that these co-residential spouse caregivers are not able to meet their health needs, thus jeopardizing their health and continuity of care. As access to healthcare is a prerequisite for the fundamental right of health [80], it is crucial to ensure that co-residential spouse caregivers have equitable access to healthcare.

To achieve this, we recommend increased economic investment in long-term care services, as well as the implementation of social and health public policies adjusted to the needs and types of informal caregivers. Since informal care in Portugal is associated with a social and family duty and a large proportion of informal caregivers do not regard themselves as such, health and social professionals should provide them with advice about their status, their rights, and the support to which they may be entitled. Formal status for Portuguese informal caregivers should also include an assessment of informal caregivers’ needs. This should lead to the provision of psychosocial support and medical support (medical appointments and regular health assessments) for co-residential spouse caregivers. Lastly, to ensure that co-residential spouse caregivers have time to rest and to promote their health, the Portuguese government should establish more effective informal caregiver respite care policies. Overall, there is a need to promote more accessible healthcare services and to implement public social and health policies that are tailored to the needs of informal caregivers in order to improve the health and healthcare use of Portuguese co-residential spousal caregivers.

## Figures and Tables

**Figure 1 ijerph-20-03975-f001:**
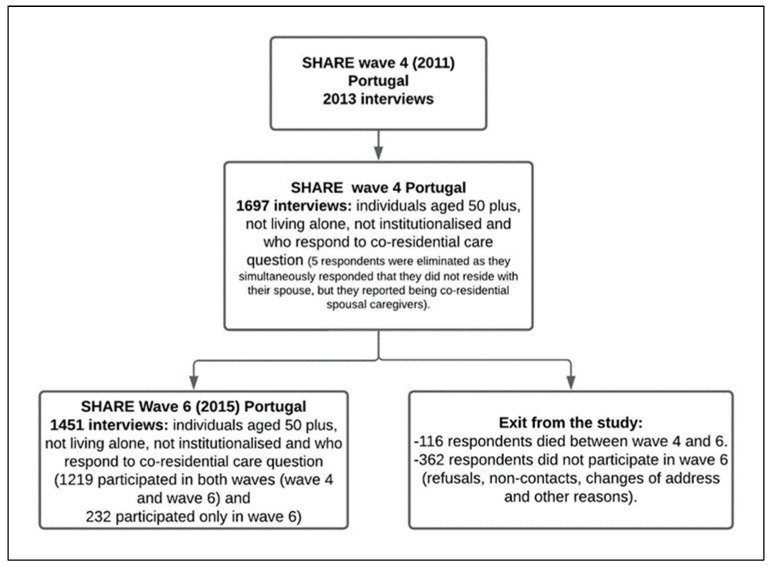
Flowchart outlining the population eligible for this study.

**Figure 2 ijerph-20-03975-f002:**
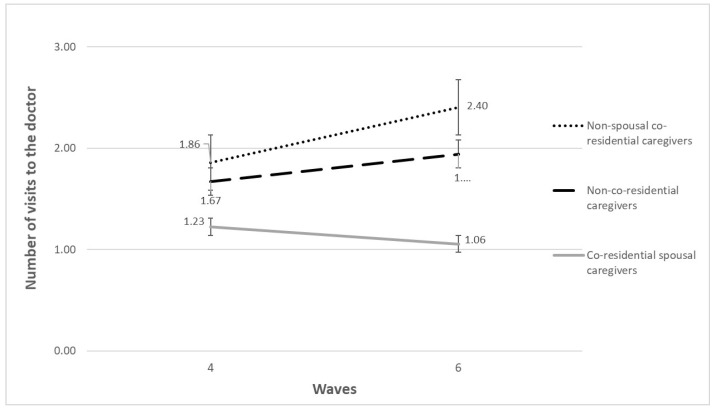
Changes in the number of visits to the doctor across waves, according to caregiver status. Source: SHARE, release 7.1.0., waves 4 and 6, unweighted data. 3142 observations and 1928 individuals. Notes: Brackets denote standard errors.

**Table 1 ijerph-20-03975-t001:** Descriptive statistics of Portuguese aged 50 plus at baseline (wave 4), according to caregiver status.

	Non-Co- Residential Caregivers (*N* = 1510)	Co-Residential Spousal Caregivers (*N* = 95)	Non-Spousal Co-Residential Caregivers (*N* = 92)	*p*-Value	*F*	X^2^
Age, mean (SD)	65.9 (10.3)	68.6 (11.4)	66.3 (11.1)	<0.001 ⋆⋆⋆	14.76	
Sex				0.004 ⋆⋆		11.13
Female	52.0%	41.2%	79.0%			
Male	48.0%	58.8%	21.0%			
Current job situation				<0.001 ⋆⋆⋆		16.67
Other situation	49.1%	24.4%	42.3%			
Retired	50.9%	75.6%	57.7%			
Education (ISCED-97)				0.466		3.58
Low education level	79.7%	81.4%	90.5%			
Medium education level	9.7%	9.8%	2.2%			
High education level	10.6%	8.8%	7.3%			
Financial distress				0.069		5.35
No	41.8%	41.4%	25.2%			
Yes	58.2%	58.6%	74.8%			
Hospitalized in the last 12 months				0.805		0.43
No	87.6%	92.5%	87.5%			
Yes	12.4%	7.5%	12.5%			
Chronic diseases, mean (SD)	1.8 (1.8)	2.3 (1.7)	2.5 (1.8)	0.011 ⋆	4.51	
Depressive symptoms (4 or more)				<0.001 ⋆⋆⋆		28.56
No	59.3%	35.7%	47.2%			
Yes	40.7%	64.3%	52.8%			
Physical inactivity				0.895		0.22
No	68.8%	76%	60.6%			
Yes	31.2%	24%	39.4%			
Household size, mean (SD)	2.7 (1.1)	2.6 (1.2)	3.2 (1.2)	<0.001 ⋆⋆⋆	11.49	
Social activities, mean (SD)	1.0 (1.3)	1.7 (1.2)	1.0 (1.3)	0.838	0.18	
Visits to the doctor in the last 12 months, mean (SD)	6.4 (14.2)	5.4 (7.5)	3.1 (3.6)	0.154	1.87	

Source: SHARE, release 7.1.0., wave 4, weighted data (percentages and means), *N* = 1697. Notes: SD = Standard Deviation; *F* = ANOVA Test; X^2^ = chi-square test; Significant associations: ´⋆⋆⋆´ < 0.001; ’⋆⋆’ < 0.01; ’⋆’ < 0.05.

**Table 2 ijerph-20-03975-t002:** Longitudinal association between co-residential care provision and healthcare use among the Portuguese population aged 50 and over.

		Null Model		Model 1		Model 2		Model 3	
		IRR (95%CI)	*p*-Value	IRR (95%CI)	*p*-Value	IRR (95%CI)	*p*-Value	IRR (95%CI)	*p*-Value
(Intercept)		3.84 (3.67–4.01)	<0.001 ⋆⋆⋆	3.78 (3.62–3.96)	<0.001 ⋆⋆⋆	1.68 (1.20–2.34)	0.002 ⋆⋆	1.69 (1.21–2.36)	0.002 ⋆⋆
Co-residential care									
	Non-co-residential caregivers			ref.		ref.		ref.	
	Co-residential spousal caregivers			1.14 (0.98–1.30)	0.092	0.98 (0.86–1.13)	0.808	1.33 (0.99–1.79)	0.061
	Non-spousal co-residential caregivers			1.04 (0.89–1.21)	0.637	1.00 (0.86–1.15)	0.746	0.89 (0.66–1.21)	0.471
Age						1.00 (1.00–1.01)	0.121	1.00 (1.00–1.01)	0.146
Sex (Male)						0.93 (0.86–1.01)	0.071	0.93 (0.86–1.01)	0.066
Current job situation (Retired)						1.04 (0.96–1.14)	0.327	1.05 (0.96–1.14)	0.325
Education (ISCED)									
	Low education level					ref.		ref.	
	Medium education level					1.02 (0.89–1.17)	0.763	1.02 (0.89–1.17)	0.748
	High education level					1.24 (1.09–1.42)	0.002 ⋆⋆	1.24 (1.09–1.42)	0.001 ⋆⋆
Financial distress (Yes)						1.07 (0.99–1.15)	0.102	1.07 (0.99–1.15)	0.099
Hospitalized in the last 12 months (Yes)						1.71 (1.56–1.89)	<0.001 ⋆⋆⋆	1.72 (1.56–1.89)	<0.001 ⋆⋆⋆
Chronic diseases						1.14 (1.12–1.16)	<0.001 ⋆⋆⋆	1.14 (1.12–1.16)	<0.001 ⋆⋆⋆
Depressive symptoms (4 or more) (Yes)						1.26 (1.17–1.35)	<0.001 ⋆⋆⋆	1.26 (1.17–1.35)	<0.001 ⋆⋆⋆
Physical inactivity (Yes)						1.15 (1.06–1.25)	0.001 ⋆⋆	1.15 (1.06–1.26)	0.001 ⋆⋆
Household size						0.97 (0.94–1.00)	0.057	0.97 (0.94–1.00)	0.052
Social activities						0.98 (0.96–1.01)	0.253	0.98 (0.95–1.01)	0.214
Time (wave)						1.07 (1.04–1.10)	<0.001 ⋆⋆⋆	1.08 (1.04–1.11)	<0.001 ⋆⋆⋆
Spouse co-residential caregiver * Time								0.86 (0.76–0.98)	0.025 ⋆
Non-spousal co-residential caregivers * Time								1.06 (0.92–1.21)	0.443
*Random Effects*									
σ (intercept)		*0.4483*		*0.4471*		*0.3127*		*0.3116*	
*Goodness-of-Fit*									
AIC		16,998.0		16,802.4		16,246.8		16,244.9	
Deviance		16,992.7		16,792.4		16,210.8		16,204.9	
Log-Likelihood		−8496.4		−8396.2		−8105.4		−8102.5	
*ICC*		*0.471*		*0.482*		*0.406*		*0.405*	
*No. of observations*		*3176*		*3147*		*3142*		*3142*	
No. of individuals		1936		1928		1928		1928	

Source: SHARE, release 7.1.0., waves 4 and 6, unweighted data. Notes: Ref. = Reference Group; IRR (Incidence Rate Ratio); 95%CI (95% Confidence Interval); AIC (Akaike Information Criterion); ICC (Intraclass Correlation Coefficient); Interaction term: ´*´; significant associations: ´⋆⋆⋆´ < 0.001; ’⋆⋆’ < 0.01; ’⋆’ < 0.05.

## Data Availability

SHARE data is available through individual user registration. All details about the application and registration process can be found at https://share-eric.eu/data/become-a-user (accessed on 22 February 2023).

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
