# Peer review of "The Longitudinal Association between Co-Residential Care Provision and Healthcare Use among the Portuguese Population Aged 50 and Over: A SHARE Study"

_ijerph, 2023, doi:10.3390/ijerph20053975_

Round 1

Reviewer 1 Report (Previous Reviewer 1)

I found the revised manuscript has improved and has a good merit on this field.

This manuscript is a resubmission of an earlier submission. The following is a list of the peer review reports and author responses from that submission.

Round 1

Reviewer 1 Report

This study has a good value to add caregiving issue. I would like to suggest that you should trim and focus co-living caregiver's burden in the introduction. The current introduction is quite long and broad. 

Also, I think you can try the model in a different way. For example, you can put one variable as "caregiver living with care recipient". You don't need to run all three models if you can design the model using the living status variable.

Also I wonder if you can consider the number of household, as a potential other caregiver. I expect you can include this variable from the main survey data. 

Reviewer 2 Report

Review Report

 1.  Originality:

The study presents interesting research carried out in a particular field. For this, the study supports the need to investigate more in this field. Similar studies have been published to which authors should better refer. It would be helpful for the authors to better describe why this paper is new and what knowledge gap it is filling.

This article needs to be revised in each section. There are often inconsistencies and the English language is misused with different grammatical errors.

Even if the abstract is unstructured, it should follow these sections: • Background: the context and purpose of the study; • Methods: how the study was performed and statistical tests used; • Results: the main findings; • Discussion; • Conclusions: brief summary and potential implications. In addition, the two last sections need to be developed and strengthened.

In general, the abstract does not express what was done in the work. Authors should rewrite it by deepening and explaining what the study actually accomplished.

2.  Relationship to Literature:

Further literature reviews (more updated) are needed to explain the relationship between the concept of "healthcare services" and “its use from co-residential care provision”.

There is what it does, and what it produces, but not what it means and who (scholar(s)) introduced this concept in the literature. The whole Introduction paragraph needs to be better explained and rephrased. To this end, it is necessary a lot of effort to delve into the literature review on the previously mentioned relationship in order to investigate what it means (from an academic point of view), and give a logical approach to the manuscript as a whole. In particular, by reading one section and then immediately the next, it seems that something has been lost in the writing. Between one section and another, it is necessary to maintain a common “thread of the discussion”. To achieve your aim, it should be necessary to perform a real review and not just explain what scholars assert. To this end, I would like to read which are the link between the theories and the real use of healthcare services among the Portuguese population published in the recent literature.

3. Methodology:

 In this section you should merge the following paragraphs and be simpler and facilitate the interpretation of the following paragraphs, would be better.

2.2. Outcome Variable

2.3. Interest Variable

2.4. Confounders

In general, a better reframed of this paragraph should be useful to better understand all the indications provided in the previous single paragraphs. For example, “Please think about your care during the last twelve months. During the last twelve months, about how many times in total have you seen or talked to a doctor about your health? Please exclude dentist visits and hospital stays but include emergency room or outpatient clinic visits. In wave 6 this question was changed slightly to: Now please think about the last twelve months. About how many times in total have you seen or talked to a medical doctor or qualified/registered nurse about your health? Please exclude dentist visits and hospital stays but include emergency room or outpatient clinic visits.”  (lines 128 - 134) if it is written in this way has no sense. It should be put differently and more clearly.

The material and method should aim to clearly show the steps you have taken to achieve your goals. More clarity is required in this section and to achieve it, the section should be revised, streamlined and simplified. I recommend referring to the methodological sections of studies already published in scientific journals. You need to make it more goal-oriented and explain what you did, without adding compound sentences that only lead to confusion.

3.                Results:

In general, the results section should follow a narrative-scientific flow to be clearer, and, also, needs to be more in-depth explained. Here there are only a few hints related to each table. No more is possible to understand. Few more sentences for each table and figure could be helpful.

4.                Discussion and Conclusions:

The implications need to be defined. Restate your topic briefly and explain why it’s important. Make sure that the discussion part is concise and clear. You should have already clear why your arguments are important in this part of your paper, and you also don’t need to support your ideas with new arguments. The list of implications should be deepened. This is a scientific article, therefore it needs to be strengthened from the point of view of academic language.

Anyway, I would suggest that a discussion section should be developed taking into account your initial research question and a clear statement of proposed contributions, once you have rephrased your arguments and developed some propositions.

In addition, in a discussion, the results obtained in the study are compared with the results obtained by other researchers, or in any case, are commented on. Here there are too many references but few comparisons are explained. As such, I suggest improving the discussion and conclusion section by adding academic and practical implications, future directions and by improving this section as a whole.
